# Plastid Phylogenomics of *Paeonia* and the Evolution of Ten Flower Types in Tree Peony

**DOI:** 10.3390/genes13122229

**Published:** 2022-11-27

**Authors:** Wen Li, Xin-Cheng Huang, Yi-Lei Wang, Rui-Ju Zhang, Dong-Yan Shi, Teng-Fei Li, Guang-Can Zhou, Jia-Yu Xue

**Affiliations:** 1College of Agricultural and Biological Engineering (College of Tree Peony), Heze University, Heze 274015, China; 2College of Horticulture, Academy for Advanced Interdisciplinary Studies, Nanjing Agricultural University, Nanjing 210095, China

**Keywords:** paeoniaceae, tree peony, chloroplast genome, codon bias, phylogenomics, flower evolution

## Abstract

*Paeonia suffruticosa* Andr., a member of Paeoniaceae, is native to China. In its 1600 years’ cultivation, more than 2000 cultivars for different purposes (ornamental, medicinal and oil use) have been inbred. However, there are still some controversies regarding the provenance of tree peony cultivars and the phylogenetic relationships between and within different cultivar groups. In this study, plastid genome sequencing was performed on 10 representative tree peony cultivars corresponding to 10 different flower types. Structure and comparative analyses of the plastid genomes showed that the total lengths of the chloroplast genome of the 10 cultivars ranged from 152,153 to 152,385 bp and encoded 84–88 protein-coding genes, 8 rRNAs and 31–40 tRNAs. The number of simple sequence repeats and interspersed repeat sequences of the 10 cultivars ranged from 65–68 and 40–42, respectively. Plastid phylogenetic relationships of *Paeonia* species/cultivars were reconstructed incorporating data from our newly sequenced plastid genomes and 15 published species, and results showed that subsect. *Vaginatae* was the closest relative to the central plains cultivar group with robust support, and that it may be involved in the formation of the group. *Paeonia ostii* was recovered as a successive sister group to this lineage. Additionally, eleven morphological characteristics of flowers were mapped to the phylogenetic skeleton to reconstruct the evolutionary trajectory of flower architecture in Paeoniaceae.

## 1. Introduction

*P. suffruticosa*, which originated in China, has been honored as the national flower of the Tang, Ming and Qing dynasties for its bright colors and elegant flowers. Andrews [1] initially completed the systematic naming of tree peony. In the following 200 years, both the Engler System and the Hooker System placed tree peony under Ranunculaceae. It was not until the 20th century that Worsdell [2] separated *Paeonia* from Ranunculaceae based on the difference in stamen arrangement between Ranunculaceae and *Paeonia*, and the new family was named Paeoniaceae [3]. To classify the species of *Paeonia* more accurately, Stern [4] further improved the classification proposed by Lynch [5] and proposed dividing the genus into sections *Onaepia*, *Paeonia* and *Moutan*. Sect. *Moutan* comprises only woody taxa (tree peony) in *Paeonia*, consisting of nine species and two subspecies that are mainly distributed in nine provinces and autonomous regions in China, including Shandong and Henan Province [6,7,8]. Sect. *Moutan* is further divided into subsections *Vaginatae* and *Delavayanae* based on the characteristics of the disc [4].

In more than 1600 years’ cultivation history, tree peony has formed seven cultivar groups worldwide, covering more than 2000 cultivars in total [9,10,11]. Four cultivar groups are located in the central plains, southwest, northwest and Jiangnan of China, and three other cultivar groups are distributed in Japan, Europe and the United States. Over the past decades, many studies have been conducted on the taxonomy, phylogenetic relationships and genetic diversity of tree peonies. With regard to the origin of tree peony cultivars, Hong [12] compared morphological characteristics and believed that the wild species *P*. *cathayana* was a close relative of *P*. *suffruticosa*, and was mainly involved in the formation of traditional tree peony cultivars in China, while Pan [13] believed that tree peony cultivars were mainly domesticated from *P*. *jishanensis*. Li [14] proposed the tree peony cultivars should have “multi-field and multi-origins”. Cheng [15] and Pei [16] considered all wild species of tree peonies to be involved in the origin of cultivated tree peony. The only difference between them was that Cheng believed that *P*. *delavayi* and *P*. *ludlowii* in subsect. *Delavayanae* were not involved in the formation of traditional tree peony cultivars in China. Cheng [15] also mentioned that *P. delavayi* and *P. lutea* were introduced to Europe and the United States at the end of the 19th century and then crossed with *P. suffruticosa* and the Japanese cultivar group to form the Lutea cultivar group. On the other hand, with the further studies using molecular markers, Meng [17], Hou [18] and Yu [19] all agreed that there was a closer kinship between cultivated tree peony and *P. ostii* and, based on randomly amplified polymorphic DNA (RAPD), amplified fragment length polymorphism (AFLP) and simple sequence repeats (SSR) markers, respectively, while there was a big disagreement concerning the kinship between other wild species and cultivated tree peony. Li [9] believed that the southwest cultivar group in China has a closer relationship with Japanese, French and American groups based on sequence-related amplified polymorphism (SRAP). In addition, Chou [20], Lin [21] and Zhao [22] reported that the DNA barcoding sequence of ITS, *Adh* and GPAT, respectively, were suitable for the study of genetic relationships among tree peony species, and these results were consistent with those based on morphological characteristics classification [23].

Flower type is one of the important ornamental characteristics of tree peony and is also an important basis for its classification. According to the ratio of diameter to height of a flower, the number, shape, size and arrangement of petals, the petaloid degree and position of stamens, and the degeneration degree of pistils, the flower types of tree peony could be divided into 10 types, including single-lobe type, lotus type, rose type, chrysanthemum type, globular type, crown type, anemone type, golden circle type, melaleuca proliferate-flower type and tower-like proliferate-flower type [24,25,26]. The natural increase in the number of petals and the centripetal petaloid of stamens formed the evolution of the melaleuca type from single-lobe type to lotus type to chrysanthemum type to rose type. In the tower-like type, the centrifugal petaloid of stamens results in the evolution from anemone type to golden circle type to crown type to globular type. Proliferate-flower types were evolved due to the massive differentiation of carpel primordia and were successively transformed and developed into various floral organs such as calyxes and petals in the melaleuca and tower-like flower types [27,28]. To date, studies on the classification and phylogenetic relationships of tree peony flower types have been rarely performed. Hosoki [29] classified 19 Chinese tree peony cultivars based on RAPD and found that only cultivars of *P. suffruticosa* cv. ‘*Shou An Hong*’ and ‘*Yao Huang*’ were consistent with Yu’s [30] classification results which combined flower types and leaf types. This may be because the designed random amplification primers did not amplify the related traits. Therefore, more molecular techniques are needed to analyze the evolutionary relationships among different flower types in tree peony.

The chloroplast is an important location for photosynthesis and a special organelle with a semi-autonomous replication ability in plant cells. With the development of high-throughput sequencing technology and the reduction of sequencing costs, the number of chloroplast genomes of angiosperms in the National Center for Biotechnology Information (NCBI) has increased exponentially. Chloroplast genomes are widely used in the study of phylogenetic relationships among orders, families, genera, species and cultivars of angiosperms [31,32,33,34,35,36] due to their characteristics of maternal inheritance, moderate rate of molecular evolution, and significant differences in the evolution of gene coding and non-coding regions [37,38,39]. In Paeoniaceae, Zhou [40] conducted phylogenetic analyses for traditional cultivars and wild species of tree peony based on 14 fast-evolved chloroplast genomic regions and 25 single-copy nuclear genes, and the results showed that the phylogeny of the wild species inferred from the nuclear genes was fully resolved and largely congruent with morphology and classification, while the chloroplast markers did not effectively resolve the interspecies relationships due to gene flow between the wild species. Moreover, by comparing the nuclear and chloroplast phylogenies, the cultivated tree peonies were proposed to have originated from homoploid hybridization among five wild species [40]. Recently, Wu [41] and Guo [42] reported that the complete chloroplast genomes showed high resolution in identification and phylogenetic analysis of the species or cultivars of tree peony. Therefore, the chloroplast genome is an ideal material for the phylogenetic study of Paeoniaceae and the 10 tree peony flower types, which have a complex genetic background and high heterozygosity. In this study, we obtained the complete chloroplast genomes of 10 representative tree peony cultivars and gained a better understanding of the evolutionary relationships among the 10 tree peony flower types through robust phylogeny.

## 2. Materials and Methods

### 2.1. Species of Paeoniaceae and Cultivars Represented by 10 Flower Types

To analyze the genetic relationship more comprehensively between the 10 tree peony flower types and species of Paeoniaceae, and to determine the evolutionary trend of tree peony in an artificial cultivation environment more systematically, we collected 15 species of Paeoniaceae with complete chloroplast genome sequences from NCBI. The 15 species were *P*. *brownii* from sect. *Onaepia*; *P*. *lactiflora*, *P*. *mairei*, *P*. *obovate* and *P*. *veitchii* from sect. *Paeonia*; *P*. *delavayi*, *P*. *lutea*, *P*. *potaninii* and *P*. *ludlowii* from subsect. *Delavayanae*, sect. *Moutan*; and *P*. *decomposita*, *P*. *jishanensis*, *P*. *ostii*, *P*. *qiui*, *P*. *rockii* and *P*. *suffruticosa* from subsect. *Vaginatae*, sect. *Moutan*. In addition, we obtained the chloroplast genome sequences of *P*. *ostii* cv. ‘*Feng Dan*’ (single-lobe type) and *P*. *suffruticosa* cv. ‘*Shou An Hong*’ (crown type) from NCBI (data from Guo et al. 2020) [42].

The other eight representative cultivars of 10 tree peony flower types were *P*. *suffruticosa* cv. ‘*Da Jin Fen*’ (lotus type), *P.×lemoinei* ‘*High Noon*’ (chrysanthemum type), *P*. *suffruticosa* cv. ‘*Da Zong Zi*’ (rose type), *P*. *suffruticosa* cv. ‘*Greendragon sleeping pool*’ (anemone type), *P*. *suffruticosa* cv. ‘*Yao Huang*’ (golden circle type), *P*. *suffruticosa* cv. ‘*Dou Lv*’ (globular type), *P*. *suffruticosa* cv. ‘*Cao Zhou Hong*’ (melaleuca proliferate-flower type) and *P*. *suffruticosa* cv. ‘*Guan Qun Fang*’ (tower-like proliferate-flower type). These cultivars, confirmed by Yi-Lei Wang, were obtained from the tree peony germplasm resources nursery of Heze Academy of Agricultural Sciences, Shandong Province, China in 2020. Additionally, these representative cultivars of 10 tree peony flower types [26] (Figure 1) are all traditional varieties and important resources for genetic breeding, possessing economic and ornamental values, e.g., different colors (white, red, green, yellow, pink, purple) and architectures of flower organs, medicinal uses and serving as a major woody oil plant species containing highly unsaturated fatty acids.

### 2.2. Sampling, DNA Extraction and Sequencing

The fresh and mature leaves of eight tree peony cultivars were sampled and frozen by nitrogen and stored in a freezer (−80 °C) for the usage of DNA extraction. A Genomic DNA Kit (TianGen Biotech CO., Beijing, China) was used to extract DNA from the leaves. The purity of the extracted DNA was tested with a NanoDrop-2000 spectrophotometer (Thermo Fisher Scientific, Wilmington, DE). Shotgun library with a 250 bp fragments interruption was built according to the manual (Vazyme Biotech Co. Ltd., Nanjing, China). After library was qualified, high-throughput sequencing with a PE150 reading length was carried out based on the Illumina novaseq 6000 platform (Annoroad Gene Technology Co., Ltd., Beijing, China).

### 2.3. Assembly and Annotation of Chloroplast Genomes

First, Trimmomatic v0.38 [43] was used to filter the adaptors and low-quality sequences in the raw data obtained by sequencing and to obtain high-quality clean reads. Subsequently, the chloroplast genome assembly software GetOrganelle v1.7.5.0 [44] was used to conduct the automatic assembly of clean reads from eight tree peony cultivars using *P*. *ludlowii* as a reference sequence under default parameters. The assembly results were then evaluated in combination with Bandage v0.8.1 [45]. Except for *P*. *suffruticosa* cvs. ‘*Cao Zhou Hong*’ and ‘*Da Jin Fen*,’ the chloroplast genomes of other cultivars showed a standard tetrad structure. Two cultivars with unloop chloroplast genome assembly results were assembled using *P*. *ludlowii* as the reference sequence under default parameters using NOVOPlasty v4.3.1 [46], and contigs of different lengths were mapped to the reference sequence using Geneious v2022.0.2 [47]. A high-quality chloroplast genome was obtained after gap filling and overlap removal. Finally, PGA software [48] was used to automatically annotate the assembled eight chloroplast genome sequences with *P*. *ludlowii* as the reference sequence, and Geneious v2022.0.2 was used to manually calibrate the annotation results. The chloroplast genomes of eight peony cultivars were uploaded to NCBI (GenBank accession number: ON243818, ON243819, ON243820, ON243821, ON243822, ON227528, ON227529, ON209192).

### 2.4. Structure, Comparison and Phylogenetic Analysis of Chloroplast Genomes

To more intuitively show the content and order of genes in the chloroplast genome, Chloroplot [49] was used to visually map the chloroplast genome. The Pfam database and TBtools [50] were then used to annotate and visualize the functional domain of the protein-coding sequences. To further determine whether there was consistency in codon usage between tree peony cultivars, CodonW v1.4.2 [51] was used to conduct codon bias analysis on 10 tree peony cultivars with default parameters. Simple sequence repeats (SSRs) were searched via MISA v1.01 [52] with the following criteria: 10, 5, 5, 5, 5 and 3 repeat units for mono-, di-, tri-, tetra-, penta- and hexa-nucleotides, respectively. Interspersed repeated sequences were searched using REPuter [53] with a Hamming distance, maximum computed repeats and minimal repeat size of 3, 80 and 30, respectively. The contraction and expansion of the four regions connected to the IR regions in the chloroplast genome were detected by IRscope [54]. mVISTA [55,56,57] was used to compare the chloroplast genomes of tree peony cultivars to detect the variation. In the process of phylogenetic tree construction, Mafft v7.453 [58] was first used to compare the whole chloroplast genomes of 15 species of Paeoniaceae and 10 tree peony flower types, and then iqtree v1.6.12 [59] was used to select the model and construct the phylogenetic tree. ML analysis was conducted with a bootstrap of 1000 repetitions.

### 2.5. Reconstruction of Ancestral Characteristics for Flower Architecture

Reconstruction for ancestral flower architectures of Paeoniaceae species was based on our highly supported phylogenetic relationships of Paeoniaceae. The morphological characteristics of flowers of Paeoniaceae species and cultivars were collected through self-observation and records from the online edition of Chinese Flora iPlant (http://www.iplant.cn/frps) (accessed on 9 May 2022) [60] (flora--Plant Species Information System; last accessed 9 May 2022) (Appendix A). All flower characteristics were further categorized into 11 traits. Mesquite V3.04 [61] was used to analyze ancestral characteristics based on the parsimony ancestral state reconstruction method used in the study of Rosaceae fruit types evolution [62].

## 3. Results

### 3.1. Basic Characteristics of the Chloroplast Genome of 10 Tree Peony Flower Types

The chloroplast genomes of 10 tree peony flower types showed a typical tetrad structure, consisting of a long single-copy region (LSC), a short single-copy region (SSC) and a pair of inverted repeat regions (IR) (Figure 2). (The chloroplast genome maps of 10 tree peony cultivars are shown in Appendix A). The chloroplast genome length ranged from 152,153 to 152,835 bp, while the genome length of the LSC, SSC and IR regions ranged from 84,213 to 85,373 bp, 17,026 to 17,056 bp and 24,863 to 25,649 bp, respectively. The GC content in the chloroplast genome ranged from 38.3 to 38.4%, and the GC content in the IR regions was significantly higher than that in the LSC and SSC regions (Table 1).

In terms of the number and order of genes in the chloroplast genome, 10 tree peony flower types were highly consistent except for *P*. *ostii* cv ‘*Feng Dan*’ and *P*. *suffruticosa* cv ‘*Shou An Hong,*’ and 136 genes were annotated. Each of the eight peony cultivars contained 88 protein-coding genes, 8 rRNAs and 40 tRNAs. Among the 136 genes, 17 contained introns, including 12 protein-coding genes and 5 tRNA genes. Most of these genes contained only one intron, while *ycf3*, *clpP* and *rps12* contained two introns, among which the *rps12* gene was a trans-splicing gene with a 5′ end in the LSC region and a 3′ end in the IR regions (Appendix A). Through the identification of the functional domains of 88 protein-coding genes, most were found to contain only one to three domains, and the *rpoB* genes had five functional domains (Appendix A).

Codon bias exists widely in natural protein translation and is influenced by many factors, including mutational pressure, natural selection and genetic drift [63,64,65]. In this study, relative synonymous codon usage (RSCU) was used to measure the codon bias of 10 tree peony cultivars because it is easy to calculate and can intuitively reflect the codon bias. The chloroplast genome contains 64 codons encoding 20 amino acids. Leucine had the most frequent codons, whereas cysteine had the least. The number of RSCU > 1 was 31, of which only UUG and UCC were G/C-ending codons, and the other 29 codons ended in A/U. Among the remaining 33 codons with RSCU ≤ 1, 30 codons ended in G/C, and only three codons (CUA, AUA and UGA) ended in A/U. The chloroplast genomes of the 10 flower types were more likely to select codons ending in A/U for amino acid synthesis (Appendix A). GC3s refers to, in addition to Met, Ser and termination codons, the frequency of G and C bases in the third codon base of synonymous codons encoding the same amino acid. The GC3s values of the 10 flower types ranged from 0.283 to 0.285, indicating that the 10 flower types had a higher bias for codons ending in A/T (Appendix A).

Interspersed repeated sequences are widely present in the chloroplast genome and include forward (F), reverse (R), complement (C) and palindromic repeat sequences (P), which repeat units with 30 bp or more. The total number of interspersed repeated sequences in the chloroplast genome of the 10 tree peony flower types was 40–44. The detected interspersed repeated sequences were forward and palindromic repeat sequences, and there were no reverse or complement repeat sequences. The lengths of the forward and palindromic repeat sequences are mainly concentrated between 30 and 60 bp. The differences between the interspersed repeated sequences of different cultivars were mainly reflected in the palindromic repeat sequences (Appendix A).

Simple sequence repeats (SSRs), also known as microsatellite DNA, are second-generation molecular markers composed of 1–6 tandem repeating nucleotide units and are often used to describe genetic differences at the individual level [66]. The total number of SSRs in the chloroplast genomes of the 10 tree peony cultivars ranged from 65 to 68. Among them, the longest tandem repeat unit was four, and no SSRs with five or six repeat units were detected. In SSRs, the number of SSRs with mononucleotides was 42–45, accounting for 61.76–69.23% of the total SSRs. There were 11–14 SSRs with dinucleotides, accounting for 16.92–20.59% of the total SSRs (Appendix A).

The IR regions of the chloroplast genome have a lower rearrangement and mutation rate than the LSC and SSC regions, which play a crucial role in the stability of the chloroplast genome [67]. Analysis of the IR regions of the chloroplast genome of 10 tree peony flower types showed that the *rps19* gene of *P*. *ostii* cv. ‘*Feng Dan*’ was completely located in the LSC region, however, the *rps19* gene of the other nine tree peony flower types was located across the LSC and IRb regions, most of which was located in LSC, and only 2 bp was located in the IRb region. The *rpl2* gene in *P*. *ostii* cv. ‘*Feng Dan*’ straddles the boundary of LSC and IRb, and the two regions are similar in length (718 bp and 778 bp, respectively). The *rpl2* gene in the other nine flower types was completely located in IRb. This may be due to the contraction of the IR region. In the SSC/IRb and SSC/IRa boundary regions, two transboundary genes, *ycf1* and *ndhF*, showed high consistency among the 10 flower types. However, the location of *trnH* genes in the LSC/IRa regions was different in different tree peony flower types; they were located in the LSC region and 3–79 bp away from the IRa region (Appendix A).

The genome comparison was based on mVISTA software using *P*. *ludlowii* of subsect. *Delavayanae* as a reference. It showed that the chloroplast genomes of the 10 tree peony cultivars selected in this experiment were relatively conserved overall, with mutations mainly occurring in some intron regions and intergenic regions and highly conserved in the protein-coding region and four rRNA (rrn16, rrn23, rrn4.5 and rrn5) regions (similarity > 90%). In terms of the four components of the chloroplast genome, the mutation rate in the IR regions was significantly lower than that in the LSC and SSC regions, indicating that the IR regions were highly conserved (Appendix A).

### 3.2. Phylogenetic Analysis of the Chloroplast Genomes of 15 Paeoniaceae Species and 10 Tree Peony Flower Types

To accurately unveil the evolutionary relationships between wild species and cultivars of Paeoniaceae, we reconstructed the phylogenetic tree (Figure 3) based on the chloroplast genomes of 15 Paeoniaceae species and 10 tree peony flower types using the maximum likelihood method (ML)**.** Our results showed that sects. *Paeonia* and *Onaepia* were each resolved as one monophyletic branch. Section *Moutan*, however, was polyphyletic, and showed a complicated composition, with subsect. *Delavayanae* containing an alien *P*.*×lemoinei* ‘*High Noon*’, and the other subsect, *Vaginatae,* inserted by *P. ostii* cv. *‘Feng Dan’* and *P. suffruticosa*. In subsect. *Delavayanae*, *P*. *lutea* was more closely related to *P*. *ludlowii* than *P*. *delavayi* and *P*. *potaninii*. In addition, *P*. *delavayi* was recovered as the sister group to *P*. *potaninii*, despite being earlier proposed as a different species by Li [27]. In subsect. *Vaginatae*, four species (*P*. *rockii*, *P*. *qiui*, *P*. *decomposita* and *P*. *jishanensis*) clustered into a monophyletic branch, with *P*. *rockii* having a closer kinship with *P*. *qiui*, and *P*. *decomposita* being more closely related to *P*. *jishanensis*. This branch did not group with the other species of the same subsection (*P*. *ostii*). *P. suffruticosa* was resolved as the sister to all other cultivars of this species and placed at the junction node of subsect. *Vaginatae* and the central plains cultivar group. Both the bootstrap support values for these were 100%.

With regard to phylogenetic relationship of 10 tree peony flower types, *P*. *ostii* cv. ‘*Feng Dan*’ (single-lobe type) and *P*. *ostii* gathered in one branch, and *P.×lemoinei* ‘*High Noon*’ (chrysanthemum type), an intrasubsectional introduction from the US, showed a closer kinship with *P*. *ludlowii* and *P*. *lutea*. The remaining eight flower types belonged to the central plains cultivar group. Among the melaleuca flower types, *P*. *suffruticosa* cv. ‘*Da Jin Fen*’ (lotus type) showed a close genetic relationship to *P*. *suffruticosa* cv. ‘*Cao Zhou Hong*’ (multi-layer proliferate-flower type), and *P*. *suffruticosa* cv. ‘*Da Zong Zi*’ (rose type) was genetically close to *P*. *suffruticosa* cv. ‘*Guan Qun Fang*’ (tower proliferate-flower type). All tower-like types clustered into a monophyletic group, with *P. suffruticosa* cv. *‘Shou An Hong’* (crown type) sister to a lineage comprising *P. suffruticosa* cv. *‘Greendragon sleeping pool’* (anemone type), *P. suffruticosa* cv ‘*Yao Huang*’ (golden circle type) and *P*. *suffruticosa* cv. ‘*Dou Lv*’ (globular type). Our results based on the plastid genomes differed from the inference of flower type evolution by Wang [28], which were based on the morphological characteristics of flower buds.

### 3.3. Reconstruction of Evolutionary Trajectory of Flower Characteristics of Paeoniaceae

Based on the robust phylogenetic relationship of Paeoniaceae and the morphological characteristics of flowers of extant Paeoniaceae species, we reconstructed the flower structure characteristics for Paeoniaceae ancestors and the evolutionary trajectory of flower architecture. (Figure 4). We inferred that the flowers of in the last common ancestor of Paeoniaceae should have two rounds of petals and that the shape of the petals was typically obovate. In terms of the number of stamens, Paeoniaceae ancestors maintained more than 100 stamens. The pistil type was apocarpous, with two to six carpels varying between individuals and smooth and hairless surfaces. In the natural evolution of the wild species of Paeoniaceae, characteristics such as petal shape, petal round number (Figure 5a), ovary position and fruit type were retained, but some distinct morphological characteristics have obviously evolved among different subgroups. One of these was the disc, a specialized receptacle in the flower between the calyx and the pistil. In the ancestors of Paeoniaceae, the receptacle between the calyx and the pistil specialized to form fleshy dentate or fleshy shallow cup-shaped discs that wrapped the base of the carpel. In later evolutions, a leathery epidermis, named a carpellary disc, appeared on the disc of the subsect. *Vaginatae*, and the disc area expanded until it completely covered the carpel; only when the petals were fully blossomed did a partially mature carpel emerge (Figure 5b). In addition, subsect. *Vaginatae* in sect. *Moutan* and some species of sect. *Paeonia* developed a dense tomentum on the carpel (Figure 5c), and the number of carpels in *P*. *ludlowii* decreased to one, in rare cases to two. This may have evolved to provide more pollination opportunities for pistils and more nutrients for fruit [62].

## 4. Discussion

Research on the origin of tree peony cultivars and the genetic relationship between cultivars has gained much help from the application of molecular technology in recent years [18,20,21,22,68]. However, studies on different molecular markers still produce controversial results due to the limitations of experimental materials and methods. In this study, the phylogenetic relationships among the 10 tree peony cultivars with different flower types and the origins of the cultivars were explored based on the data of complete chloroplast genomes. In terms of phylogenetic relationships among cultivars, *P.×lemoinei ‘High Noon’* (Lutea cultivar group), *P. ostii* cv *‘Feng Dan’* (Jiangnan cultivar group) and the other 8 cultivars (central plains cultivar group) were each grouped into one clade according to the classification of cultivar groups. These results indicate that there were genetic discrepancies among different cultivar groups due to different origins from different ancestral species. In terms of the origin of cultivars, the phylogenetic topology showed that the tree peony model plant *P. suffruticosa* was more closely related to the central plains cultivar group, which was consistent with the suggestion by Hong [12] that *P. suffruticosa* was involved in the formation of traditional Chinese tree peony cultivars. Further tracing of the ancestors of the central plains cultivar group revealed that wild species of subsect. *Vaginatae* were more closely related to the central plains cultivar group. This is consistent with the inference by Meng and Zhang [17], who studied the relationship between wild tree peony species and cultivars based on RAPD, and suggested that wild species of subsect. *Vaginatae* participated in the formation of the original cultivar groups. However, Wang [69], Li [14] and Yu [70] proposed that, based on evidence from morphology and cytology, *P. jishanensis*, *P. rockii* and *P. ostii* were mainly involved in the formation of the central plains cultivar group, and our study further proposed that, compared with *P. jishanensis*, *P. ostii* might be more genetically involved according to the phylogeny. Meng [17], Hou [18] and Han [71] obtained the same results as this study using RAPD, AFLP and SRAP markers, respectively. However, there are still some discrepancies between the chloroplast genome and the above molecular markers with regard to the relationships of *P. decomposita* and *P. qiui* to the central plain cultivar group. It is speculated that the chloroplast genome is maternally inherited, and there is a preference for parental selection in the cultivation process of tree peony cultivars [72], resulting in discordance between the chloroplast genome and nuclear markers.

The emergence of simple and double flower traits in tree peony is a significant feature [27], and an obvious sign of flower type artificial evolution, probably under strong and directional selection. Additionally, attention has been paid to other ornamental characteristics of tree peony, such as petal color and shape in the process of artificial breeding; thus, there is no obvious evolution in disc shape, carpel number or other characteristics compared with wild species. However, the numbers of petals and petal rounds have evolved rapidly. Based on the features of flower bud differentiation, Wang [28] proposed the evolutionary forms of melaleuca and tower-like flower types, which provided a reference for predicting the variation amplitude of flower types and the characteristics of further evolution. Among the 10 tree peony flower types in this study, the melaleuca and tower-like flower types each clustered into one branch, indicating that cultivars with different evolutionary degrees of the same flower type probably had similar evolutionary relationships, which has important reference significance for selecting parents for directional breeding of flower types. However, the evolutionary sequence within the same flower type was somewhat different from Wang [28], which may be related to the differences in the ancestor species and flower color of the 10 tree peony flower types [73]. Therefore, cultivars with a common ancestor and different flower types of the same color system can be selected for further study.

In the long-term evolution process of Paeoniaceae, the tree peony flower types gradually diversified and complicated due to the extensive homoploid hybridization involved in tree peony domestication [24,40]. The flower types of tree peony could be divided into 10 types based on the ratio of diameter to height of a flower, the number, shape, size and arrangement of petals, the petaloid degree and position of stamens and the degeneration degree of pistils [24,25]. Among these variable factors, the petaloid of stamens and degeneration of pistils are the main reasons that affect the diversity of flower types in tree peony. Moreover, the phenomena of centripetal and centrifugal petaloid of stamens are common in tree peony [26], whereas only centripetal stamen petaloid exists in some other plants, such as the formation of double flowers in *Malus halliana* [74], rose [75], *Prunus mume* [76] *Actaea rubra* [77] and *Camellia japonica* [78]. The centrifugal petaloid of stamens provides the basis for the formation of the rich floral types of tree peonies, as the case of golden-circle type tree peony with stamens between the normal petals and petaloid stamens. Similarly, the peony, a herb perennial taxa and sister to tree peony in *Paeonia*, had the diversified and complicated flower types formed by centripetal and centrifugal petaloid of stamens [79]. Gao [80] speculated that the evolution mechanism of tree peony flower type accorded with the ABC model of floral development in angiosperm species [81]. Recently, studies of genes associated with the petaloid of stamens had been performed in some species, and showed that the homeotic ABC genes encode members of the MADS-domain class of transcription factors regulating the petaloid of stamens in *Camellia changii* [77], *Paeonia lactiflora* [82] and *Nelumbo necifera* [83]. According to the ABC model, it was inferred that the mechanism of gene over-expression and homologous heterologous inheritance were the main regulatory mechanisms for the diversity of flower types in tree peony.

In this study, based on the phylogenetic relationship between wild species of Paeoniaceae and 10 tree peony flower types, we reconstructed their ancestral characteristics to gain a deeper understanding of the evolutionary relationship between wild species and cultivars of Paeoniaceae. However, further studies are needed to explore the deep network evolution between cultivars and ancestral species and the molecular evolutionary relationship between different flower types. A systematic understanding of the evolutionary relationship between wild species and high-value cultivars of tree peony is not only beneficial to the utilization of rich genetic resources of wild species and traditional cultivars, but also has important guiding significance for individualized directional breeding.

## Figures and Tables

**Figure 1 genes-13-02229-f001:**
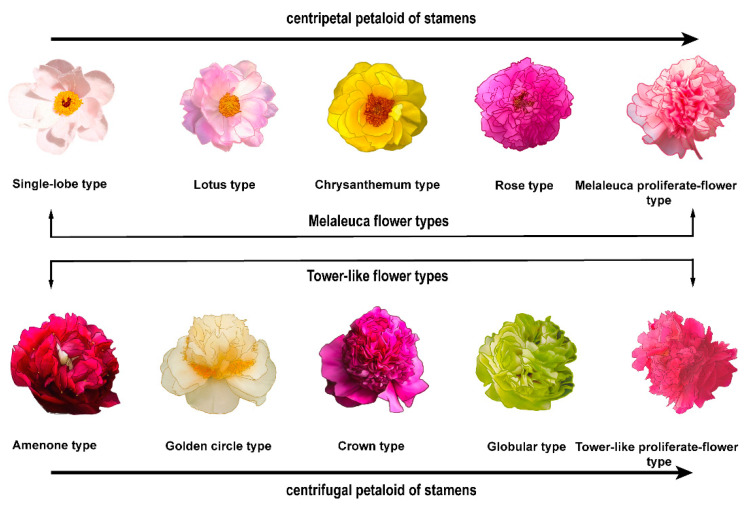
Flower types of all 10 tree peony cultivars. From left to right, from top to bottom were *P*. *ostii* cv. ‘*Feng Dan*’ (single-lobe type), *P*. *suffruticosa* cv. ‘*Da Jin Fen*’ (lotus type), *P.×lemoinei* ‘*High Noon*’ (chrysanthemum type), *P*. *suffruticosa* cv. ‘*Da Zong Zi*’ (rose type), *P*. *suffruticosa* cv. ‘*Cao Zhou Hong*’ (melaleuca proliferate-flower type), *P*. *suffruticosa* cv. ‘*Greendragon sleeping pool*’ (anemone type), *P*. *suffruticosa* cv. ‘*Yao Huang*’ (golden circle type), *P*. *suffruticosa* cv. ‘*Shou An Hong*’ (crown type), *P*. *suffruticosa* cv. ‘*Dou Lv*’ (globular type) and *P*. *suffruticosa* cv. ‘*Guan Qun Fang*’ (tower-like proliferate-flower type).

**Figure 2 genes-13-02229-f002:**
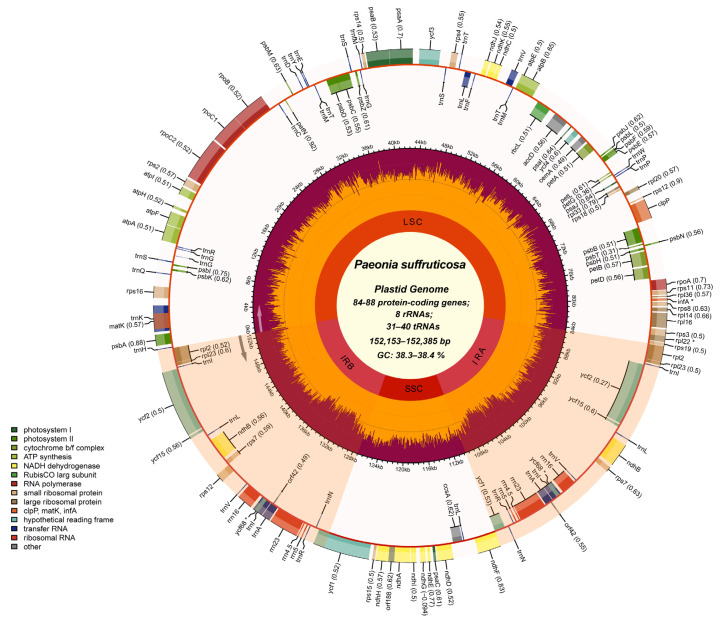
Chloroplast genome map of 10 tree peony flower types using *P. suffruticosa* cv. *‘Da Zong Zi’* as the template. The gradient GC content of the genome was plotted in the second circle, with a zero level based on the outer circle. The gene names and their codon usage biases were labeled on the outermost layer. An * in the name of a gene indicates that the gene is a pseudogene. The gene-specific GC content was depicted with the proportion of shaded areas. Represented with arrows, the transcription directions for the inner and outer genes are listed clockwise and anticlockwise, respectively. (The chloroplast genome maps of the 10 tree peony cultivars are shown in Appendix A).

**Figure 3 genes-13-02229-f003:**
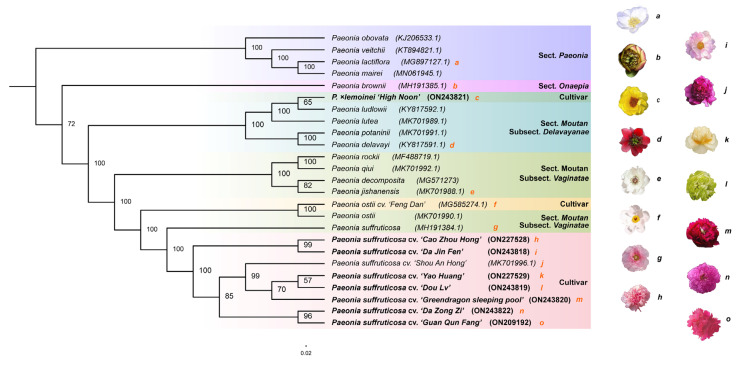
Phylogenetic tree constructed using the maximum likelihood (ML) method based on the complete chloroplast genome sequences of 15 Paeoniaceae species and 10 tree peony flower types. (**a**). *P. lactiflora*; (**b**). *P. brownii*; (**c**). *P*.*×lemoinei* ‘*High Noon*’; (**d**). *P. delavayi*; (**e**). *P. jishanensis*, (**f**). *P. ostii* cv. ‘*Feng Dan*’; (**g**). *P. suffruticosa*; (**h***). P. suffruticosa* cv. ‘*Cao Zhou Hong,*’ (**i**). *P. suffruticosa* cv. ‘*Da Jin Fen*’; (**j**). *P. suffruticosa* cv. ‘*Shou An Hong*’; (**k**). *P. suffruticosa* cv. ‘*Yao Huang*’; (**l**). *P. suffruticosa* cv. ‘*Dou Lv*’; (**m**). *P. suffruticosa* cv. ‘*Greendragon sleeping poor*’; (**n**). *P. suffruticosa* cv. ‘*Da Zong Zi*’; (**o**). *P. suffruticosa* cv. ‘*Guan Qun Fang*.’ Among the 25 chloroplast genome sequences for phylogenetic tree construction, species (cultivar) names in bold were taken from self-sequenced data, whereas those not in bold were downloaded from the NCBI database.

**Figure 4 genes-13-02229-f004:**
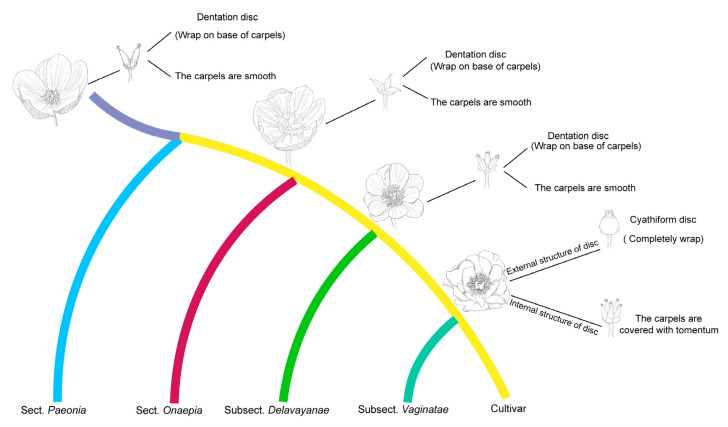
Evolutionary histories of flower types in Paeoniaceae. The evolutionary history of different flower types in Paeoniaceae was reconstructed based on Paeoniaceae phylogenetic relationship and 11 morphological characteristics. Based on the reconstruction results of ancestral characteristics, we predicted the representative flower types of the four evolutionary node species of Paeoniaceae. The evolution of floral organs mainly focuses on the shape and number of petals, the characteristics of disc and the number of carpels.

**Figure 5 genes-13-02229-f005:**
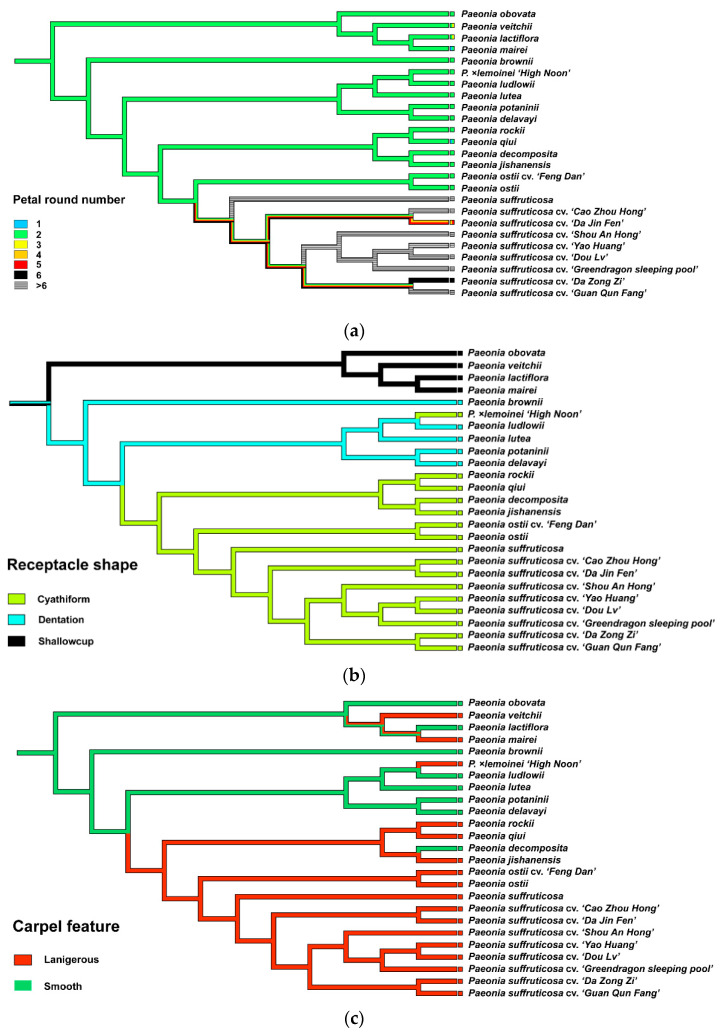
Reconstruction of ancestral characteristics in the context of the topology in Figure 2. The color in the square represents the specific characteristics of this morphological trait in the species (cultivars). The colors of different nodes in the topology represent the inference results of the corresponding ancestral characteristics based on the morphological characteristics of the species (cultivars) and the topology of Figure 2. (The other six reconstruction results of the ancestral characteristics of Paeoniaceae based on morphological features and topological structure are located in Appendix A). (**a**) Reconstruction of ancestral petal round numbers in the context of the topology in Figure 2; (**b**) Reconstruction of ancestral receptacle shape in the context of the topology in Figure 2; (**c**) Reconstruction of the ancestral carpel feature in the context of the topology in Figure 2.

**Table 1 genes-13-02229-t001:** Basic information of chloroplast genomes of ten flower types.

Cultivar	All Length	GC%	LSC Length	GC%	SSC Length	GC%	IR Length	GC%	GenBank Accession Numbers
*P. ostii* cv. *‘Feng Dan’*	152,153 bp	38.3	85,373 bp	36.7	17,054 bp	32.7	24,863 bp	43.1	MG585274.1
*P. suffruticosa* cv. *‘Da Jin Fen’*	152,835 bp	38.3	84,485 bp	36.6	17,056 bp	32.6	25,647 bp	43.1	ON243818
*P.×lemoine ‘High Noon’*	152,519 bp	38.4	84,213 bp	36.7	17,026 bp	32.8	25,640 bp	43.1	ON243821
*P. suffruticosa* cv. *‘Da Zong Zi’*	152,820 bp	38.4	84,471 bp	36.6	17,051 bp	32.6	25,649 bp	43.1	ON243822
*P. suffruticosa* cv. *‘Greendragon sleeping pool’*	152,820 bp	38.4	84,475 bp	36.6	17,051 bp	32.6	25,647 bp	43.1	ON243820
*P. suffruticosa* cv. *‘Yao Huang’*	152,819 bp	38.4	84,474 bp	36.6	17,051 bp	32.6	25,647 bp	43.1	ON227529
*P. suffruticosa* cv. *‘Shou An Hong’*	152,819 bp	38.4	84,474 bp	36.6	17,051 bp	32.6	25,647 bp	43.1	MK701996.1
*P. suffruticosa* cv. *‘Dou Lv’*	152,819 bp	38.4	84,474 bp	36.6	17,051 bp	32.6	25,647 bp	43.1	ON243819
*P. suffruticosa* cv. *‘Cao Zhou Hong’*	152,835 bp	38.3	84,485 bp	36.6	17,056 bp	32.6	25,647 bp	43.1	ON227528
*P. suffruticosa* cv. *‘Guan Qun Fang’*	152,820 bp	38.4	84,471 bp	36.6	17,053 bp	32.6	25,649 bp	43.1	ON209192

## Data Availability

The datasets generated for this study can be found in the National Center for Biotechnology Information (NCBI) (https://www.ncbi.nlm.nih.gov/, accessed on 17 October 2022) and obtained from GenBank under accession (numbers ON209192, ON227528, ON227529, ON243818, ON243819, ON243820, ON243821, and ON243822).

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
