# Peer review of "Plastid Phylogenomics of Paeonia and the Evolution of Ten Flower Types in Tree Peony"

_genes, 2022, doi:10.3390/genes13122229_

Round 1

Reviewer 1 Report

Manuscript Number: genes-2010471-peer-review-v1

In this manuscript entitled “Plastid Phylogenomics of Paeonia and the Evolution of Ten Flower Types in Tree Peony”, the authors sequenced plastid genome of 10 representative tree peony cultivars corresponding to 10 different flower types. Subsequently, phylogenetic relationships were explored with 15 published species in Paeoniaceae showed that subsect. After systematic analysis, the authors showed that P. ostii was a sister group to this cultivar group, and inferred the evolutionary trajectory of flower architecture in Paeoniaceae. However, I think the manuscript needs some English editing to improve the language. But in my opinion, there are many avoidable mistakes in the manuscript, and the author should carefully check and correct them. Below are some detailed remarks from my side.

Abstract

Line 24: The "and" is in the wrong font.

Introduction

Line 69: What are the results of Yu [20] et al? This sentence is not clear. Please rewrite it “Hosoki [19] classified 19 Chinese tree peony cultivars based on random amplified polymorphic DNA (RAPD) and found that the results were not completely consistent with Yu [20].”

Line 59-71: Some studies have developed other molecular markers or other methods for the same genus. I think important to mention these studies as antecedent in the introduction.

Line 73-88: Some studies have reported chloroplast genomes for the same genus, but the author doesn't mention it at al. I think important to mention these studies as antecedent in the introduction.

Line 83-87: 10 tree peony flower types, what are they?

Materials and Methods

Line 90-107: Please add floral photos of these materials. In addition, there are many varieties of tree peony. Why did the author choose these varieties as representative? What is the basis?

Line 98-99: “In addition, we obtained the chloroplast genome sequences of P. ostii cv. ‘Feng Dan’ (single-lobe type) and P. suffruticosa cv. ‘Shou An Hong’ (crown type) from NCBI.” Please added the references.

Line 101-107: The background of these plant materials taken from the Academy of Agricultural Sciences should be briefly introduced.

Line 112: “The fragments of qualified DNA were……”, what is the standard of qualified?

Line 154-155: Please added the website of the online edition of Chinese Flora iPlant.

In this part, please add manufacturers and countries for used reagents/ instrument. Such as: a NanoDrop-2000 spectrophotometer.

Results

Line 263-273: In Figure2, there are 25 species (cultivar), but only 15 species (cultivar) showed flower type pictures, added the flower type pictures of the other 10 species (cultivar). In addition,

Line 274-287: By what method were the patterns of these peony varieties determined? Such as: P. suffruticosa cv. ‘Da Zong Zi’ (rose type).

Line 288-308: The order of Figures 3 and 4 should be reversed. In addition, how to quantify the “indefinite” traits among the 11 morphological characteristics?

Discussion

In this part, the authors said cultivars with different evolutionary degrees of the same flower type had similar evolutionary relationships. I want to know, There are many varieties of peony, and many flower types are in transition state. Is it reliable to explore the evolutionary relationship of flower types with only 10 flower types? In addition, the authors should compare their results with those obtained from studies on the evolution of flower patterns in peonies or other woody plants.

Reviewer 2 Report

The authors tried to solve the controversies regarding the origin of tree peony cultivars and the phylogenetic relationships between different cultivar groups. They assembled some plastid genome and performed comparative analysis of representative Paeoniaceae species including 10 representative tree peony cultivars corresponding to 10 different flower types.

This is an important study to find the origin of the tree paeony cultivars.

Nevertheless, the following issues should be revised.

Lines 4 and 429,

There is no description for the contribution of some authors, Yi-Lei Wang, Rui-Ju Zhang, Dong-Yan Shi, and Teng-Fei Li.

Line 263,

Please give a description for the values on each branch.

Line 331,

Please give a name of the figure. 

Line 396.

In Supplementary Materials, please correct the table numbers.

Round 2

Reviewer 1 Report

This paper would be more suitable for publication in a related article on chloroplast genome sequencing.